# Fully Human Antibodies for Malignant Pleural Mesothelioma Targeting

**DOI:** 10.3390/cancers12040915

**Published:** 2020-04-08

**Authors:** Fabio Nicolini, Martine Bocchini, Davide Angeli, Giuseppe Bronte, Angelo Delmonte, Lucio Crinò, Massimiliano Mazza

**Affiliations:** 1Biosciences Laboratory, Istituto Scientifico Romagnolo per lo Studio e la Cura dei Tumori (IRST) IRCCS, 47014 Meldola, Italy; fabio.nicolini@irst.emr.it (F.N.); martine.bocchini@irst.emr.it (M.B.); 2Unit of Biostatistics and Clinical Trials, Istituto Scientifico Romagnolo per lo Studio e la Cura dei Tumori (IRST) IRCCS, 47014 Meldola, Italy; davide.angeli@irst.emr.it; 3Department of Medical Oncology, Istituto Scientifico Romagnolo per lo Studio e la Cura dei Tumori (IRST) IRCCS, 47014 Meldola, Italy; giuseppe.bronte@irst.emr.it (G.B.); angelo.delmonte@irst.emr.it (A.D.); lucio.crino@irst.emr.it (L.C.)

**Keywords:** malignant pleural mesothelioma (MPM), immunotherapy, fully human antibody, tertiary lymphoid structure (TLS), therapeutic antibody, MPM management, mesothelioma, solid tumors targeting, BCR repertoire, phage display

## Abstract

Immunotherapy is the most promising therapeutic approach against malignant pleural mesothelioma (MPM). Despite technological progress, the number of targetable antigens or specific antibodies is limited, thus hindering the full potential of recent therapeutic interventions. All possibilities of finding new targeting molecules must be exploited. The specificity of targeting is guaranteed by the use of monoclonal antibodies, while fully human antibodies are preferred, as they are functional and generate no neutralizing antibodies. The aim of this review is to appraise the latest advances in screening methods dedicated to the identification and harnessing of fully human antibodies. The scope of identifying useful molecules proceeds along two avenues, i.e., through the antigen-first or binding-first approaches. The first relies on screening human antibody libraries or plasma from immunized transgenic mice or humans to isolate binders to specific antigens. The latter takes advantage of specific binding to tumor cells of antibodies present in phage display libraries or in responders’ plasma samples without prior knowledge of the antigens. Additionally, next-generation sequencing analysis of B-cell receptor repertoire pre- and post-therapy in memory B-cells from responders allows for the identification of clones expanded and matured upon treatment. Human antibodies identified can be subsequently reformatted to generate a plethora of therapeutics like antibody-drug conjugates, immunotoxins, and advanced cell-therapeutics such as chimeric antigen receptor-transduced T-cells.

## 1. Introduction

Malignant pleural mesothelioma (MPM) is an aggressive neoplasm with a dismal prognosis, median overall survival (OS) of 14 months, whose onset is associated with asbestos exposure [1]. Europe currently carries most of the global asbestos-related disease burden due to heavy asbestos use during earlier decades [2]. A peak in MPM incidence is expected in the 2020s due to the long disease latency, although most countries have already banned asbestos use [1,3,4,5]. In contrast, countries that still employ asbestos are very likely to display a substantial burden of asbestos-related disease and MPM nowadays and in the future. 

About 60% of MPM patients carry mutations in the BRCA1 associated protein-1 (*BAP1*) gene [6]. BAP1 is a protein involved in DNA repair mechanisms, cell cycle control, apoptosis, and carcinogenesis [7,8,9,10,11,12,13]. BAP1 mutational status is clearly associated with the insurgence of MPM [9,10,12,13,14,15], the response to chemotherapy [16], patient survival [17], and, when coupled to other DNA repair gene alterations, it has suggested synthetic lethality therapeutic approaches [18,19]. MPM is frequently diagnosed at a late stage due to the lack of early symptomatology, reliable biomarkers, and screening tools. Current therapies in clinical practice consist of surgery, radiotherapy, and chemotherapy. We recently reviewed all current and innovative therapeutic approaches and most relevant clinical trial results [20]. Such a detailed survey revealed that new therapeutic modalities and prognostic biomarkers are urgently needed in order to grant a fair chance of survival to all MPM patients. In this review, we examine the current unmet clinical needs in MPM, concentrating on immunotherapy dilemmas, highlighting the main emerging experimental therapies and clinical evidence, and, above all, exploring and analyzing the approaches that can be used to identify new fully human MPM-targeting antibodies for future therapies. To our knowledge, no other review has tackled the issue of targeting MPM from the same angle and perspective, providing useful suggestions regarding novel technologies to achieve this goal.

## 2. Immunotherapy in MPM

Immune checkpoints (IC) proteins, such as cytotoxic T-lymphocyte-associated antigen 4 (CTLA-4), programmed death 1 (PD-1) and PD-L1, are immune system regulators that maintain homeostasis and prevent autoimmunity in physiological conditions [21]. Their overexpression in MPM keeps the anti-tumor immune response in check, creating a local immunosuppressive microenvironment [22,23]. IC inhibitors (ICIs), i.e., antibodies targeting CTLA-4, PD-1, or PD-L1, are used as immunomodulatory agents to interfere with the CTLA-4:B7-1/2 or PD-1:PD-L1 axes and to help to overcome tumor-immune escape [24,25,26] with very different efficacy. 

Recently, PD-L1 expression in malignant mesothelioma has been assessed on tissue microarrays using two different FDA-approved antibodies, and 22% to 27% of cases were positive for PD-L1 (1% cut off) [27]. PD-L1 is expressed by a substantial proportion of biphasic and sarcomatoid MPM cases, and its positivity above 1% is associated with a significant 10-months reduction in median OS compared to PD-L1 negative tumors [28,29]. Similarly, high PD-L1 expression (>50%) in epithelioid MPM patients correlates with shorter PFS (6.7 vs. 9.9 months) [28]. Despite its prognostic value [30,31,32], PD-L1 expression is not a valid predictive marker of response to anti-PD-L1 therapies for several tumor types [33,34], including MPM [35]. Anti-PD-1/PD-L1 therapies were tested in different trials as second or third-line treatment in MPM patients [36,37,38,39,40,41,42,43], but, to date, only nivolumab has obtained regulatory approval in chemo-refractory mesothelioma patients in Japan [44]. At present, the pembrolizumab plus platinum/pemetrexed-based chemotherapy (PPC) combination as first-line treatment, in comparison to pembrolizumab or PPC alone, is being evaluated in the phase III trial NCT0278417, while nivolumab is being investigated in the randomized phase III trial CONFIRM (NCT03063450) in comparison with placebo [41]. The activity of durvalumab, a PD-L1 inhibitor, in combination with first-line PPC, was tested in the DREAM study (ANZ clinical trial registry number: ACTRN12616001170415). This treatment resulted in an objective response rate (ORR) of 61% using mRECIST and 53% using iRECIST criteria and in progression-free survival at 6 months of 71% (mRECIST). On the basis of these observations, a randomized phase 3 trial will be started [45]. 

ICs expression is variable among tumor cells and strictly controlled at different stages of T-lymphocyte activation. For these reasons, a combination strategy employing two different ICIs in addition to chemotherapy has been proposed to achieve a synergistic effect by overcoming immune-resistance observed in some MPM patients. Encouraging results observed for different ICIs in combination [46,47,48] prompted the investigation of the nivolumab plus ipilimumab combination in comparison to standard PPC alone as a first-line option in the phase III clinical trial Checkmate-743 (NCT02899299). 

At present, efficacy and safety of adoptive T-cell therapies, in particular chimeric antigen receptor-transduced T-cells (CAR-T), in MPM and other solid tumors are under investigation [49,50]. CAR-T-cells directed against mesothelin (MSLN), a glycoprotein expressed on MPM and other solid tumor cells, with a limited presence on normal tissues [51], represent a promising therapeutic option and many efforts have been made to improve their clinical efficacy and safety profile [52,53]. Recently, Adusumilli and colleagues reported the outcome of a phase I clinical trial, NCT02414269, [54,55] on patients with MPM and pleural metastases from lung or breast cancer treated with anti-MSLN CAR-T-cells. Of note, the inclusion of anti-PD-1 therapy was essential to elicit clinical efficacy and to avoid T-cell exhaustion since no patient had an objective response before pembrolizumab addition, showing the importance of also modulating the immune suppressive features of the tumor microenvironment (TME) in this therapeutic setting. The pembrolizumab plus anti-MSLN CAR-T-cell combination results in the best clinical outcome with an ORR of 63% (10/16) and a disease control rate of 75% (12/16). No evidence of on-target, off-tumor, or therapy-related toxicities higher than grade 1 was observed. Although applied to a limited number of patients so far, CAR-T therapies against MPM have shown impressive results, highlighting the difference in efficacy for advanced cell therapies compared to small molecule drugs or antibodies. Recently, a comprehensive review of immunotherapy in MPM has been published [56]. However, it is evident that the limited availability of therapeutically targetable antigens hinders the effectiveness of these strategies in MPM patients and this issue will need to be addressed in the future.

## 3. Making a Hot Tumor Microenvironment 

The efficacy of ICIs in MPM patients highlights the presence and the activity of immune cells in situ able to fight cancer if properly unleashed. However, to achieve this goal, TME must be modified in order to abolish/interfere with specific immune-suppressive cues. Interestingly, Barsky and colleagues recently reported a case of a man with MPM treated with a combination of palliative radiation and immune-gene therapy (GM-CSF) [57]. The outcome of this treatment combination was outstanding, resulting in a so-called “abscopal effect”. In oncology, the abscopal effect is a phenomenon by which localized radiation induces an anti-tumor response at distant sites. RT can trigger an immunogenic cell death (ICD) [58,59] and can stimulate antigen-specific, adaptive immunity [60]. ICD leads to subsequent anti-tumor immune responses, including the release of tumor antigens by irradiated tumor cells, the cross-presentation of tumor-derived antigens to T-cells by antigen-presenting cells (APCs), and the migration of effector T-cells from the lymph nodes to distant tumor sites suggesting that irradiated tumors can act as an in situ vaccine if appropriate conditions are in place [61,62,63]. The overall incidence of the abscopal effect of RT alone is low, with 46 clinical cases reported from 1969 [63]. Those statistics may be explained by the insufficiency of RT alone to overcome the immunoresistance of malignant tumors. Given that immunotherapy can reduce host immune tolerance towards tumors, it is possible that the combination of RT and immunotherapy can amplify the anti-tumor immune response, a hypothesis currently under investigation within the phase I trial, NCT02959463, in which adjuvant pembrolizumab after RT in lung-intact MPM patients is under evaluation. In a murine model of MPM, the abscopal effect can be induced by local RT and enhanced by immune-suppressive CTLA-4 blockade as infiltrated T-cells, both in primary and secondary tumor sites, are predominantly composed by anti-tumor cytotoxic CD8+ T-cells, while immunosuppressive regulatory T-cell (Tregs) are reduced [64]. Those observations corroborate the idea that systemic tumor response can be induced or unleashed by modifying the TME features locally.

## 4. Tertiary Lymphoid Structures in Solid Tumors and MPM: Where the Anti-Tumor Response Begins

Efficient adaptive responses against cancer occur typically in secondary lymphoid organs (SLOs), wherein major histocompatibility complex (MHC) molecule–peptide complexes are presented by dendritic cells (DCs) to T-cells, and require the migration of DCs from the tumor site to the SLOs [65]. 

B-cells are activated in the SLOs upon antigen binding in primary follicles and receive help from the CD4+ T-cells to proliferate and form a secondary follicle that will become a germinal center (GC). This process allows lymphocyte proliferation and differentiation into effector T-cells and memory B-cells (MBCs) that migrate into the tumor, leading to its destruction if no antagonizing and exhaustive cues are in place. 

However, studies on the immune context of tumors revealed that anti-tumor defenses can also occur at the tumor site within organized lymphoid aggregates resembling SLOs [66] called tertiary lymphoid structures (TLSs) [67]. TLSs are also found in the stroma at the invasive margin and/or core of different tumor types [68,69]. TLSs are composed of a T-cell-rich zone harboring mature DCs juxtaposed to a B-cell follicle with GC characteristics surrounded by plasma cells (PCs). TLSs are privileged sites for local presentation of neighboring tumor antigens to T-cells by DCs and activation, proliferation, and differentiation of T and B-cells, resulting in the generation of effector memory T helper (TH) cells and effector memory cytotoxic cells, MBCs, and antibody-producing PCs [70,71,72,73,74,75]. TLS density is associated with higher numbers of CD8+ and CD4+ T-cells in tumors [76], and evidence indicates that TLSs play a major role in controlling tumor invasion and metastasis. A positive impact of TLS density on OS and disease-free survival in lung cancer [71,76,77,78], colorectal cancer [79,80], pancreatic cancer [81,82], oral squamous cell carcinoma [83], and invasive breast cancer [70,84,85,86] has been documented. Importantly, its prognostic value is independent of tumor–node–metastasis (TNM) staging in most malignancies, suggesting TLS can induce a systemic, long-lasting anti-tumor response. High endothelial venules (HEVs) similar to those that allow entry of lymphocytes into SLOs are usually detected near TLSs [70]. In this context, HEVs allow lymphocytes to enter into tumors. Therefore, therapies enhancing this feature would be beneficial to improve anti-tumor immune responses. Tregs negatively affect HEV formation, and their depletion in murine models of cancer lead to increased T-cell infiltration and activation, and to the eradication of the disease [87,88]. Tregs and other immune-suppressive cell types, such as myeloid-derived suppressor cells (MDSCs), regulatory B-cell (Bregs), and soluble factors like TGFb and IL-10, contribute to the development of an immune-suppressive TME. Tumor-resident Tregs co-express high levels of CTLA-4, OX-40, and GITR compared to effector T-cells, and in murine models of mesothelioma, the combination of anti-OX-40 and anti-CTLA-4 has a synergistic effect and results into a 20% to 80% increase in tumor regression as compared to single-antibody treatment [89]. Coherently with this picture, the combination of anti-angiogenic drugs with anti-PD-L1 therapies increases HEV and TLS formation in murine models of breast cancer and neuroendocrine pancreatic tumors [90], supporting the idea that a powerful anti-tumor systemic response by ICIs is sustained, if not triggered, by the presence of TLSs in situ. TLS heterogeneity among human cancers, analyzed via a pan-cancer gene expression analysis of TME cellular composition on The Cancer Genome Atlas (TCGA) data and MPM, as well as lung adenocarcinoma and lung squamous cell carcinoma, displays a high expression of a 12-chemokine gene signature associated with TLS composition [91], hinting at a frequent, but also heterogeneous, presence of TLSs [92].

In MPM, lymphoid aggregates are present in about 70% of tumors, and GCs within these aggregates can be spotted in about 30% of cases [23]. These aggregates show functional similarity to TLSs, in which T- and B-lymphocytes are apart in two adjacent regions surrounded by HEV, as already shown in ovarian and prostate cancer [93,94]. Despite that, clear evidence of HEV’s presence in MPM is still lacking. Importantly, the presence of intratumoral CD8+ T-lymphocytes is an independent good prognostic marker associated with a reduced risk of death for MPM patients [73]. 

Additionally, structural inter- or intra-chromosomal rearrangements and single nucleotide variants have been recently reported from mate-pair and RNA sequencing-based analyses on mesothelioma specimens predicting the expression of potentially-targetable neoantigens [95]. Moreover, some of these neoantigens bind patient-specific MHC, and specific tumor-infiltrating T-cell clones are expanded as observed through TCR repertoire analysis [95]. In line with these observations, TCR diversity and mutation/neoantigen load have been inversely correlated, but both active and suppressive TME immune components, such as Treg and CD8+ T-cells, were present and equally balanced suggesting a scenario where activated anti-tumor CD8+ T-cells are counteracted by pro-tumoral immune suppressive molecules and Treg cells [96] or activated CD8+ T and CD4+ T-helper cells displaying phenotypic markers of exhaustion including PD-1, TIM-3, and LAG3 positivity [30]. 

## 5. The Importance of B-cell Infiltration in Solid Tumors and MPM

B-cell follicles in TLS from non-small cell lung cancer and ovarian cancers contain bona fide Ki67+ GC B-cells expressing the activation-induced deaminase (AID) gene, encoding an enzyme critical for somatic hypermutation and class switch recombination of immunoglobulin genes, as well as, of BCL-6, the transcription factor involved in the late stage of B-cell maturation [71,97]. Additionally, the presence of CD38+ CD138+ PCs around the follicle is highly suggestive of antibody production in situ [98]. Indeed, micro-dissected follicles subjected to BCR repertoire analysis revealed clonal amplification compared to peripheral B-cells, suggesting a local antigen-driven B-cell response to several malignancies [93,97,99,100,101,102]. Additionally, PCs isolated from dense aggregates in tumor stroma [98], produce anti-tumor antibodies of the immunoglobulin G (IgG) isotype in vivo, whose mechanism of action has not been yet determined. One possibility is that anti-tumor IgGs produce locally increase antigen presentation by DCs and/or directly promote the activity of specific subsets of CD4+ T-cells endowed with Fcγ receptors (FcγRs) [25]. In favor of this scenario is the presence of IgG deposits in TLS, the spatial organization of TLSs that may favor DC priming by locally produced IgGs, and the observation that tumor-derived immune complexes increase the expression of the co-stimulatory molecule CD86 on DCs in vivo [93]. In favor of the latter are the results of a meta-analysis in a large set of human cancers showing that the prognostic effect of T-cells is generally stronger when tumor-infiltrating B-cells or PCs are present, highlighting the importance of the coordination between cellular and humoral adaptive immune responses [103]. 

The role of B-cells and the association of B-cell rich TLSs with survival and anti-PD-1 immunotherapy response in sarcoma and melanoma have been recently established [104,105]. Importantly B-cells are the strongest prognostic factor even in the context of low CD8+ T-cells [104] in sarcoma, and class-switched MBCs are specifically enriched in melanoma ICI-treated responders [106].

In murine models of mesothelioma treated with immunotherapy, the presence of B-cells is essential for a good prognosis, indicating that antibodies are generated and contribute significantly and essentially to the therapeutic effect [107]. Consistently, B-lymphocyte infiltration in MPM tissue positively correlates with prognosis [108] although variable in its extent [109]. Moreover, clinical [110] and preclinical data on B-lymphocytes’ contribution to MPM prognosis suggest that they elicit an adaptive cytotoxic immune response rather than acting directly as antigen-presenting cells (APCs) [107,111]. In this respect, MPM shares many similarities with other solid tumors previously described and provides a unique opportunity to develop novel immunotherapies and to identify novel MPM targeting receptors in patients.

## 6. The Quest for Specificity in Malignant Mesothelioma: How Do We Fill This Gap?

Adoptive cell therapies, in combination with ICIs, have shown very promising results in MPM. The specificity or preference of targeting is granted almost exclusively by the use of antibodies or their derived fragments directed to tumor-specific/associated antigens. First, attempts of therapy using murine monoclonal antibodies (mAbs) in cancer patients failed due to the rapid generation of neutralizing antibodies and because of a mismatch with components of the human immune system. These results showed the importance of using human or human-compatible/tolerable biomolecules and prompted the design of novel screening platforms to find them. 

Both antigen-based (Figure 1) and -unbiased screening methods (Figure 2) can be used to this end. The first approach is based on previous knowledge of tumor-specific/associated antigens located on cell membranes, whereas the second leverages on the possibility to test a priori the binding of antibodies to cells restricting the successive identification of antigens to lead candidates only.

## 7. Phage Display Screening Using Human Synthetic Antibody Libraries

Next-generation sequencing applied to tumor cells is a powerful tool to identify and quantify expressed neoantigens as mutations/fusions or as alternative splicing variants [112]. Frequency data from RNA-Seq datasets can inform on tumor heterogeneity and guide patient stratification. In lung cancer patients, for example, the abundance of alternative splicing variants correlates with the longest OS [113], suggesting that variants can be displayed and targeted by adaptive immunity. Altered sequences generated by improper splicing or mutations can be expressed and used to generate human antibodies specific to these moieties. A convenient strategy to generate human antibody fragments (Fabs and scFvs) against specific antigens or cells is phage display (reviewed in [114]). The importance of phage display technology was restated in 2018 by the award of Nobel Prize in Chemistry to George P. Smith and Sir Gregory P. Winter ”for the phage display of peptides and antibodies”. Phage display has allowed the production of clinically relevant antibodies (reviewed in [115]) via the classical approach that relies on the incubation of antibody-displaying phages with an antigen (biopanning), either produced as a purified protein immobilized on a solid substrate or expressed on a host cell surface for consecutive rounds of phage library and antigen exposure to progressively decrease antibody library complexity in favor of specificity (see Figure 1). While the first approach is straightforward and usually employs proteins expressed in bacteria or mammalian cells to recapitulate potential post-translational modifications, the latter allows the expression of the antigen in its physiological context, e.g., on cell membrane, at the cost of increased complexity of screening setup. Classically, single bacterial clones are selected and grown to produce antibodies or phages displaying specific antibodies and test their binding to the target of interest individually. 

Phage display can also be used to discover antibodies binding to cancer cells rather than to a specific antigen. Biopanning on cancer cells aims to find antibodies able to bind to cancer-specific and not-yet-characterized antigens (see Figure 2). Nowadays, next-generation sequencing provides an efficient, quantitative and fast way to analyze the evolution of complexity of phage antibody-display libraries during consecutive biopanning enrichment stages in order to predict putative antigen or tumor cell binders subsequently produced, reformatted, and tested for their affinity to the target of interest [116].

An unbiased phage display approach has been used to identify mesothelioma tumor-targeting scFvs that recognize antigens expressed on mesothelioma cells and tissues of both sarcomatoid and epithelioid histotype. In this study, 95 mesothelioma-targeting scFvs were identified, and 21 candidates were further characterized by FACS, IHC, and for their in vitro internalization capacity on mesothelioma cells with the goal to deliver conjugated anti-tumor drugs directly inside tumor cells [117]. Further analyses identified MCAM/CD146, a surface transmembrane glycoprotein with adhesion functions that belongs to the immunoglobulin superfamily, as one of the antigens. CD146 had been previously described as a marker for advanced melanoma [118] and other tumors [119,120]; it is expressed by all mesothelioma histotypes and by a limited spectrum of normal human adult tissues, among that some vascular endothelial and smooth muscle cells [121]. Currently, the utility of MCAM/CD146 detection in pleural effusion fluid and peripheral blood samples is being investigated as a diagnostic and prognostic marker for MPM [122]. The generation of a phage antibody-display library from the entire antibody genes repertoire of a cancer patient has also been attempted. Rare cancer-targeting antibodies have been identified by this strategy [123]. However, the immunodominance phenomenon typical of certain cancers [114,124,125] has hindered a wider use of this strategy in early attempts.

## 8. Fully-Human Antibodies from Genetically Modified Transgenic Mice

Besides phage display, an ever more popular approach to generate human antibodies deploys humanized mice engineered to express human immunoglobulin chains. Several models have been generated over the past 25 years by different engineering strategies (reviewed in [126]). These transgenic animals can be vaccinated with human antigens, and single B-cell clones or derived hybridomas can be screened for the production of targeting antibodies via ELISA or flow cytometry-based assays. Companies like TRIANNI Inc., Kymab Limited [127], and Creative Biolabs [128] developed proprietary animal models to pursue this strategy. These platforms feed internal pipelines with novel biologics and are available to academic and private institutions in a pay-for-service modality or through the licensing of animals. Advantages include generation in vivo of high affinity, matured IgG antibodies (preferred format for therapy) bypassing additional affinity maturation steps typically used in phage display, high efficiency of the system and low hands-on-time required, ease of automation and throughput of screening. However, this strategy is not suited to raise antibodies against antigens extremely conserved between human and mouse since the murine immune system counter selects self-antigens to prevent the generation of autoantibodies. Human synthetic antibody phage display libraries, on the contrary, can be used to raise antibodies against any moiety at the cost of longer subsequent optimization steps. Both technologies have contributed to the generation of clinically relevant antibodies in several fields [129]. 

## 9. From Today’s Patients the Future Cures for MPM 

As explained above, patients develop an immune response against MPM that, if unleashed, can be very effective. The presence of TLSs and the insurgence of oligoclones of B-cells inside or at the border of MPM tissue are positive prognostic features and constitute a window of opportunity to capture human therapeutic antibodies. Now the next question is, how can we exploit this powerful reservoir of biologics to isolate or design novel targeting drugs? In other words: what technologies are available to take up this challenge?

## 10. BCR Repertoire from Sequencing Data

Bulk RNA-Seq data from tumor tissue contain a hitherto overlooked picture of a tumor and its ecosystem. Typically, this data is used to evaluate the expression of known transcripts, while de novo formed sequences, like those generated by T- and B-cells in the assembly and generation of their specific receptors, are usually discarded since they cannot be easily matched with the reference transcriptome. However, these sequences can be retrieved from raw data and employed to extract the sequence of TCRs and BCRs from tumor tissue infiltrated immune cells using specific bioinformatic tools. One of them is MiXCR [130], a universal tool that takes raw sequencing data, including RNA-Seq, as input and profiles TCR and BCR repertoires. As a reference, it uses a built-in library of V, D, J, and C gene sequences from human or mouse. As an output, it provides a list of clonotypes derived by assembling identical and homologous reads, corrected for sequencing errors. V’DJer is another framework that can be applied to RNA-Seq data for this purpose [131]. It can be run on BCR light and heavy chain data and employs unmapped paired-end short reads aligning them against a reference transcriptome. Then, V’DJer detects VDJ rearrangements, generates contigs, and quantifies the ones that represent the most abundant portions of the BCR repertoire. When the expression levels of BCR are low, there is an option to increase the sensitivity of the algorithm at the cost of increasing the demand for computational resources. V’DJer has been used, for example, to retrieve antibodies from RNA sequencing data of melanoma patients from TCGA repository [131,132]. At present, TCGA contains expression analyses of 87 MPM patients (TCGA-MESO) that could be used for this purpose. In addition, RNA can be obtained from FFPE samples specifically showing TLSs by IHC in prospective and retrospective cohorts of patients. It is possible to infer the sequence of resident B-cell clones by applying bioinformatic tools to RNA-Seq or by sequencing amplicons for immunoglobulin chains using specific sets of degenerate universal primers from whole tissue DNA or RNA/cDNA. The latter approach is implemented by the immunoSEQ platform (Adaptive Biotechnologies, Seattle, WA). In contrast to profiling using bulk RNA-Seq data, it is more precise since the experimental design is optimized to identify the BCR repertoire through the ImmunoSeq Analyzer software, specifically for this data. Its starting material can be both genomic DNA (gDNA) and cDNA: in order to assess the clonal expansion of B-cells in tissues, gDNA is the best solution since each cell contains the same copy number, while mRNA transcripts can be very different among cells, depending on cellular activation and even the retrotranscription procedure can add other confounding factors. However, cDNA is a better starting material when the goal is to study different antibody abundance since there is a difference in the mRNA expression between activated and naive B-cells. Finally, independently of the method employed for their derivation, identified immunoglobulin heavy and light chain sequences can be further used to build and produce candidate antibodies to test their ability to bind to MPM target cells. 

## 11. MPM Patient Memory B-cell Receptor Repertoire Exploitation to Find Novel Therapeutic Antibodies

A second powerful approach to obtain human antibodies targeting MPM cancer cells directly exploits the immune system of patients. Individuals exposed to viral agents, parasites, and tumors develop an adaptive response against non-self and neoantigens. Anti-cancer treatments such as vaccines and ICIs elicit impressive clinical responses (reviewed in [25]) and immunological memory in subgroups of cancer patients (“elite responders”). MPM is not characterized by a high mutational burden [15], an important determinant of the response to checkpoint blockade.

The efficacy of the anti-PD-1 pembrolizumab was shown by Alley and colleagues in KEYNOTE-028 [38]. Additionally, treatment with the ICI ipilimumab in combination with anti-TGF-β and anti-CD25 antibodies of syngeneic MPM in BALB/c mice resulted in (i) disease eradication in all treated animals, (ii) elevated levels of tumor-specific IgG antibodies in cured mice, (iii) failure to regrow tumors in cured mice when re-challenged with the same tumor, and (iv) response abolition in the absence of B-cells, suggesting that antibodies generated upon treatment contribute significantly to the curative effect [107]. Besides that, CD20+ B-cells infiltration in MPM tumor tissue has a positive prognostic value, as previously discussed [108].

Therefore, the immune system of elite responders can be mined to isolate MBCs producing targeting antibodies. MBCs derive mostly from affinity matured and somatic hypermutated B-cells from the germinal centers [133] and constitute a reservoir of high-affinity antibody producers. These features make the MBC pool very attractive so that companies invest in the design of screening platforms to exploit it. For example, Oncoresponse, a company that developed a proprietary, clinically validated human-antibody discovery platform in partnership with MD Anderson Cancer Center, follows this paradigm and aims to identify therapeutically relevant antibodies from patients showing elite response against cancer after immunotherapy. MBCs are easily accessible from the peripheral blood of donors and are suitable for viral immortalization to generate lymphoblastoid cultures for high throughput screens. MBC immortalization involves the infection and transformation of peripheral MBCs by Epstein Barr Virus (EBV) [134] or by BCL-6 / BCL-XL expressing vectors [135] and generates cells that express BCR on the membrane and release their antibody into culture medium at the same time. BCR presence is exploited to isolate cells binding to labeled soluble antigens by cell sorting [135] so that subsequently immunoglobulin sequences from isolated cells can be cloned into expression vectors for large-scale antibody production. Companies like Humabs and AIMM therapeutics exploit those strategies to raise antibodies against specific targets. However, the same technology can be used to isolate targeting antibodies in an antigen-unbiased manner, as shown for melanoma via cell-based screenings of EBV immortalized B-cells [136]. Additionally, human plasmablasts and MBCs can also be cultured for a limited time using specific cytokines [136,137,138,139,140,141]. Importantly, these approaches do not rely on prior knowledge of a specific target. Instead, target identification is postponed, initially drawing on the demonstration of efficacy and specificity towards MPM cancer cells. MBCs receptor repertoire can also be obtained by RNA-Seq from peripheral blood or draining lymph node purified MBCs mining for de novo formed or highly enriched variants after treatment in elite responders [142]. The advantages and drawbacks of the different screening strategies for fully human antibody selection are summarized in Table 1.

## 12. Conclusions

Despite amazing efforts made by the scientific and medical community and the plethora of therapeutic options developed over the last decades, the discovery of a curative MPM treatment remains elusive and is an unmet clinical need. To date, the most promising therapeutic approaches comprise immunotherapies and CAR-based therapies that have shown impressive although preliminary clinical achievements. The necessity of discovering novel antigens and ways to target them to cope with tumor heterogeneity and to provide more effective combined treatments for patients is now clear, and future therapies cannot disregard it. The most innovative screening technologies for the generation of fully human antibodies are in place and combine elements from fields of science that started far apart and came together to serve the purpose. These include protein engineering, next-generation sequencing (NGS), virology, cell biology, and genetic modeling of animals, providing an opportunity to find novel and unknown therapeutic targets for MPM and cancer in general. Based on these premises, we believe that a future breakthrough in MPM management will come from the design of adoptive cell therapies engineered to target antigens that are still unknown, but that can be identified via unbiased screening strategies.

## Figures and Tables

**Figure 1 cancers-12-00915-f001:**
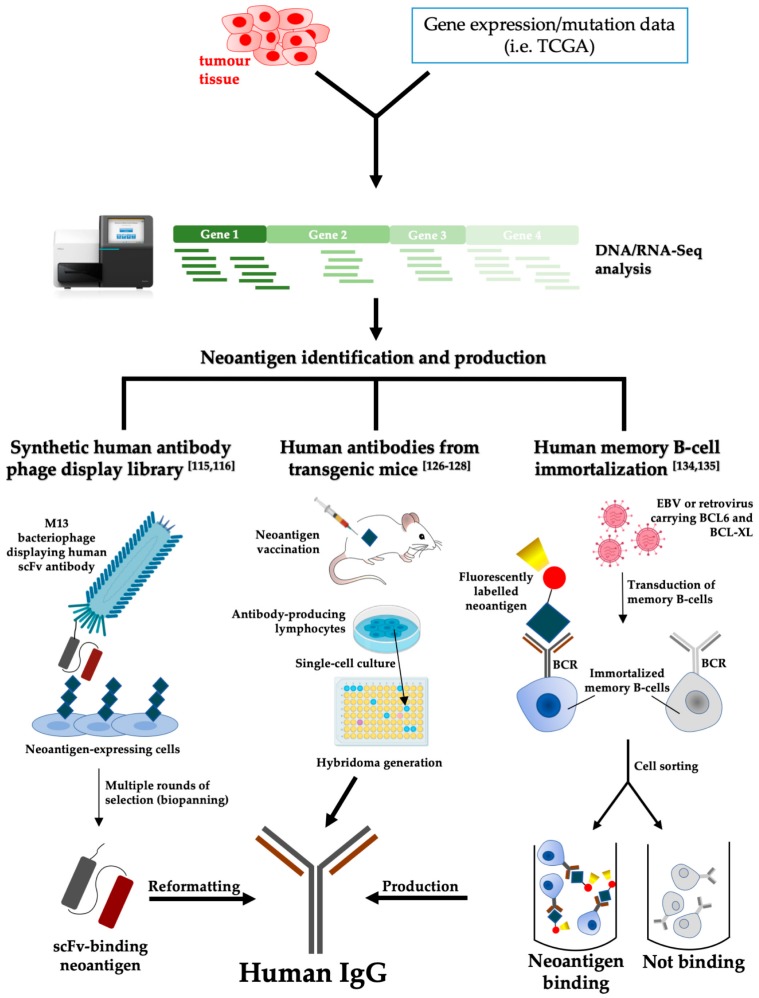
Schematic representation of antigen-based screening strategies to obtain fully human tumor-targeting antibodies.

**Figure 2 cancers-12-00915-f002:**
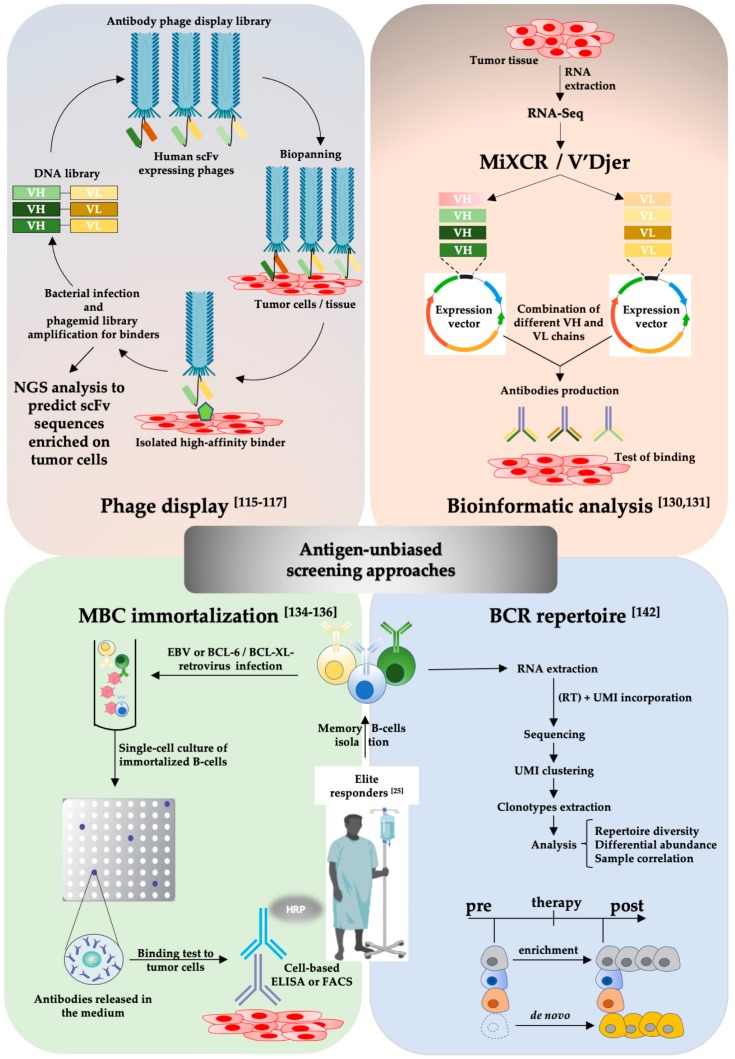
Schematic representation of 4 antigen-unbiased screening strategies to obtain fully human tumor-targeting antibodies.

**Table 1 cancers-12-00915-t001:** Advantages and drawbacks for all screening strategies used to obtain fully human antibodies.

Screening Strategy	Approach	Antigen Display	Advantages	Disadvantages
Antigen-based screening (pre-selection of specificity based on antigen expression)	Phage-display technology with human antibody synthetic libraries [115,116]	In vitro adsorbed antigen	Does not require expensive instrumentation Applicable to any moiety anchored on a substrate or exposed on cells (biopanning) Established protocols Fastest strategy to lead candidates Analysis of library complexity and prediction of binders via NGS	Fab or scFv fragment production (affinity maturation steps often needed) Requires reformatting to IgG format
		Expression of the antigen on the host cell membrane (biopanning)		
	Transgenic mice expressing fully human antibodies [126,127,128]	Vaccination with soluble antigen	Quick (3–4 months turn around time to lead candidates) Fully human antibodies in IgG format Established protocols No requirement for affinity maturation steps	Requires an animal facility Expensive More difficult to apply it for plasma membrane antigens
	MBC immortalization via BCL6/BCL-XL expression [134,135]	Soluble and fluorescently labeled antigen	Ease of blood samples collection from elite responders or volunteers In vivo affinity matured human immunoglobulins	Requires a BSL2 area Requires the sorting of very rare populations Expensive instrumentation (cell sorter) Requires the rescue of VH and VL IgG chain sequences at early stages (AID expression) Requires the production of labeled-antigens
Antigen-unbiased screening (selection based on binding to cancer cells)	Phage-display technology with human antibody synthetic libraries [115,116,117]	Antigen on cell surface	Does not require expensive instrumentation Applicable to any cell type Established protocols Fastest strategy to lead candidates Analysis of library complexity and prediction of binders via NGS	Fab or scFv fragment production (affinity maturation steps often needed) Requires reformatting in IgG format Requires a test of binding specificity to normal human tissues *a posteriori*
	BCR repertoire from the peripheral blood of elite responders pre- and post-therapy [142]	Antigen on cell surface	Ease of blood sample collection from elite responders In vivo affinity matured human immunoglobulins	Possible downsampling Requires cloning and production of the antibodies VH and VL pairs are not known (unless single-cell sequencing is used) Requires a test of binding specificity to normal human tissues a posteriori
	Bioinformatic analysis of BCR repertoire in tumor tissue [130,131]	Antigen on cell surface	Availability of large number of FFPE samples Applicable to retrospective case series Applicable to any RNA-Seq dataset	Requires cloning and production of the antibodies Possible downsampling due to low quality or limited sample material VH and VL pairs cannot be known Requires a test of binding specificity to normal human tissues a posteriori
	EBV infection [134]	Antigen on cell surface	Easy availability of elite responder samples (blood/PBMCs) Established protocols Isolation of in vivo high-affinity matured and human-compatible immunoglobulins Basic technical expertise on viral manipulation	Requires a BSL2 area Identification of the antigens can be technically challenging Requires a test of binding specificity to normal human tissues a posteriori

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
