# Peer review of "Fully Human Antibodies for Malignant Pleural Mesothelioma Targeting"

_cancers, 2020, doi:10.3390/cancers12040915_

Round 1

Reviewer 1 Report

Many thanks to the authors for revising their article according to the comments. I am happy with the revised manuscript and therefore approve it for publication.

Author Response

Point 1. Many thanks to the authors for revising their article according to the comments. I am happy with the revised manuscript and therefore approve it for publication.

Response 1. We thank reviewer 1 for the appreciation.

Reviewer 2 Report

I read this manuscript with great interest as well as the reponses to the reviewers comments. I think that all comments and points raised by the reveiwers have been addressed. The manuscript has been considerably improved both in content and form. From my point of view, this manuscript is acceptable for publication and can be published as it is. 

Author Response

Point 1. I read this manuscript with great interest as well as the reponses to the reviewers comments. I think that all comments and points raised by the reveiwers have been addressed. The manuscript has been considerably improved both in content and form. From my point of view, this manuscript is acceptable for publication and can be published as it is.

Response 1. We thank reviewer 2 for the revision and comments.

Reviewer 3 Report

The authors have addressed many of the concerns that had been raised by the reviewers and as a result the manuscript has improved; however, there is still room for improvement. For example, the abstract continues to suffer from a lack of coherence and hasn’t been revised appropriately. The abstract requires a substantial rewriting to reflect the purpose and objectives of the work and summarize the content of the paper. The authors state that a comprehensive review on immunotherapy in MPM has been published elsewhere (Ref: Front. Oncol. 2020, 10, 187). Therefore, I recommend that, in the Introduction, the authors explicitly explain what distinguishes this review from previous publications on this topic; hence the rationale for the present work.

Other points:

In page 3: Several acronyms have been used without proper definition; i.e., PPC, CCP, ORR, DCR, SLO, TME.  All acronyms need to be clearly spelled out the first time that they appear in the paper.

Page 3, sentence starting in line 2: “At present” and “currently” serve the same purpose. Remove one of them from the sentence. 

Author Response

Point 1. The authors have addressed many of the concerns that had been raised by the reviewers and as a result the manuscript has improved; however, there is still room for improvement. For example, the abstract continues to suffer from a lack of coherence and hasn’t been revised appropriately. The abstract requires a substantial rewriting to reflect the purpose and objectives of the work and summarize the content of the paper.

Response 1. We thank reviewer 3 for appropriated comments and suggestions. Following reviewer 3 suggestion we have completely rewritten the abstract in order to be more coherent with the rest of manuscript remarking the purpose and aims of the work.

Point 2. The authors state that a comprehensive review on immunotherapy in MPM has been published elsewhere (Ref: Front. Oncol. 2020, 10, 187). Therefore, I recommend that, in the Introduction, the authors explicitly explain what distinguishes this review from previous publications on this topic; hence the rationale for the present work.

Response 2. The previous work (Nicolini et al. Front. Oncol. 2019, 9, 1519 [20]) was a comprehensive review about all current and innovative therapeutic approaches for MPM, including (but not only) immunotherapy. In this work, we underline the importance to identify targeting molecules for MPM, suggesting different approaches to identify new antibodies. To our knowledge no other review has tackled with the same angle and perspective the issue of targeting MPM, providing useful suggestions regarding novel technologies to achieve this goal. We explicit this concept in lines 91-97.

Other points:

In page 3: Several acronyms have been used without proper definition; i.e., PPC, CCP, ORR, DCR, SLO, TME.  All acronyms need to be clearly spelled out the first time that they appear in the paper.

Yes, we revised all acronyms as suggested. See lines 123, 128, 129, 157, 199 and 153, respectively.

Page 3, sentence starting in line 2: “At present” and “currently” serve the same purpose. Remove one of them from the sentence. 

Thank you, ‘’currently’’ has been removed (see line 125).

This manuscript is a resubmission of an earlier submission. The following is a list of the peer review reports and author responses from that submission.

Round 1

Reviewer 1 Report

The review "Malignant Mesothelioma:the quest for specificity" aims to discuss methods allowing identification of novel human antibodies and their utilisation for development of more efficient treatments in this disease. 

This review could be informative, if the authors decided to focus on the specific issue of identification and utilisation of antibodies from start to end of the article. Instead, the current text is a long, non-cohesive and confusing account of mesothelioma current treatment options, current clinical trials, only to reach the real scope of the review in the last couple of chapters. My main comments are listed below:

The abstract is very vague and confusing. A review of language and layout is suggested. Line 37: reference needed. Line 38: "peak in mesothelioma incidence"- is this referring to Italian, European or global incidence? Line 40 it is mentioned that mesothelioma is diagnosed late. This is wrong- usually it is late stage disease by the time of symptoms.  Line 41: resistance to treatment as part of mesothelioma biology needs to be mentioned. The introduction is not representative of the main text that follows. There is a whole chapter on radiotherapy, but not one on chemotherapy. Why? Tertiary lymphoid structures in solid tumours and MPM. Very small part of this chapter is dedicated to mesothelioma. Too much detail about the biology, what happens in other cancers and the authors' personal views. Lines 163-165: references needed. Lines 165-167: more recent references exist. Lines 227-308: please list all past and current clinical trials in mesothelioma, as well as preclinical CAR T cell targets Language and layout need to be revised.

Reviewer 2 Report

I've read the manuscript on quest for specificity in malignant mesothelioma with great interest. The manuscript is basically a review article on immune responses in association with current and future immunotherapy in malignant pleural mesothelioma.

Manuscript starts with the description of the current therapy for MPM, then follows the comprehensive review of different aspects of immune responses, current trials from immunotherapy, MPM specific markers as potential targets for immunotherapy and the methods for further improvements and discoveries in this field. The relevant literature is included and properly quoted.The discussion at the end of manuscript is relevant and not too long. List of the literature covers all important publications.

From the point of view of the review article, this manuscript is acceptable for publication after correction of some minor language inaccuracies.

Reviewer 3 Report

The authors are trying to provide an overview the status of the rapidly evolving field of immunotherapy in MPM which is a big undertaking. In the current form I do not recommend the publication of the review. However, if the below points are all adequately addressed then this summary can be helpful to give a current update on this very hot issue.

Major general points:

In general a large number of citations are referring to other tumors and it is often not clear from the text which statements are mesothelioma associated. Please revise accordingly. In contrast important mesothelioma studies are sometimes missing, details provided below. Figure 1, 2 and Table 1 has absolutely no mesothelioma specific information. It would only be useful for mesothelioma researchers if MPM references or specifics are included in the figures. Table 1 should definitely somehow indicate and list which approaches were already attempted in mesothelioma with references given. English must be improved here and there in the manuscript (e.g. abstract first sentence…). Overall the abstract is hard to follow, partly due to the language.

Major specific issues:

The title is misleading; the review is about various immunotherapy related approaches in MPM. In the light of our quest for specificity, please provide a more scientific title. Also since the authors focus on pleural mesothelioma it should be stated in the title..

There are already several studies regarding PD-L1 in MPM, the authors must refer to them and provide range of incidence rather than stating substantial. The complement system is completely omitted from the review; despite the fact that it plays an important role both in mesothelioma and in immune response; please discuss this aspect (e.g. PMID 31681580, 31649675, 29209316, 29184132). The tertiary (or often called ectopic) lymphoid structures are discussed almost exclusively in non-MPM tumors (with an unnecessary high number of non-MPM citations), please review our current knowledge about them in MPM instead. The authors provide no papers that demonstrate the presence/relevance of HEVs in MPM. Are there any studies? The authors state here that it was not studied ”in detail”. If there is no evidence it should be mentioned as such. A couple of reviews already addressed the role of immunotherapy in MPM, at least a few major one should be cited. Important preclinical MPM studies are overlooked and missing (e.g. PMID 25980578) while plenty of non-MPM work is cited. The nintedanib studies are missing from the anti-angiogenesis overview. The phage display text is absolutely general, completely lacks specifics and fails to refer to any of MPM phage display studies!!! Some MPM oncolytic virus studies are also not cited. Same for CAR T-cell therapy… While the elite responder concept is intriguing the authors should briefly discuss whether elite-responder MPM patients have been or can be identified and how?

Minor points:

Line 38 Asbestos is still in use and ubiquitously present, the sentence is misleading.

Line 62 should refer to original paper PMID 28687356

Some citations in the bibliography need to be corrected (e.g. ref 16, ref 73)